# A cytidine deaminase regulates axon regeneration by modulating the functions of the *Caenorhabditis elegans* HGF/plasminogen family protein SVH-1

**Tatsuhiro Shimizu, Takafumi Nomachi, Kunihiro Matsumoto** ⓘ *, **Naoki Hisamoto** *

Division of Biological Science, Graduate School of Science, Nagoya University, Chikusa-ku, Nagoya, Japan

* g44177a@nucc.cc.nagoya-u.ac.jp (KM); i45556a@cc.nagoya-u.ac.jp (NH)

**Data Availability Statement:** All data generated or analyzed during this study are included in this article (and its supplementary information files).

## Abstract

The pathway for axon regeneration in *Caenorhabditis elegans* is activated by SVH-1, a growth factor belonging to the HGF/plasminogen family. SVH-1 is a dual-function factor that acts as an HGF-like growth factor to promote axon regeneration and as a protease to regulate early development. It is important to understand how SVH-1 is converted from a protease to a growth factor for axon regeneration. In this study, we demonstrate that cytidine deaminase (CDD) SVH-17/CDD-2 plays a role in the functional conversion of SVH-1. We find that the codon exchange of His-755 to Tyr in the Asp–His–Ser catalytic triad of SVH-1 can suppress the *cdd-2* defect in axon regeneration. Furthermore, the stem hairpin structure around the His-755 site in *svh-1* mRNA is required for the activation of axon regeneration by SVH-1. These results suggest that CDD-2 promotes axon regeneration by transforming the function of SVH-1 from a protease to a growth factor through modification of *svh-1* mRNA.

## Author summary

The axon regeneration pathway in *C. elegans* is activated by SVH-1, a growth factor that belongs to the HGF/plasminogen family. SVH-1 is a dual-functional factor that promotes axon regeneration as an HGF-like growth factor and regulates development as a plasminogen-like protease. It is crucial to understand the mechanism by which SVH-1 transforms from a protease to a growth factor during axon regeneration. In this study, we demonstrate that CDD-2, a cytidine deaminase, is critical for converting SVH-1 function from a protease to a growth factor by modifying *svh-1* mRNA.

## Introduction

The ability of neurons to regenerate after injury is determined by a complex interplay between their intrinsic growth program and external cues [1]. Injured neurons need to receive accurate and timely information about the extent of axonal damage to successfully increase their

**Funding:** This work was supported by grants from the Japan Society for the Promotion of Science (https://www.jsps.go.jp/english/e-grants/index.html, JP21H02578 to N.H. and JP19H00979 to K.M.) and the Project for Elucidating and Controlling Mechanisms of Aging and Longevity from Japan Agency for Medical Research and Development (https://www.amed.go.jp/en/index.html, JP21gm5010001 to N.H.). T.S. received both research funding and salary from a Japan Society for the Promotion of Science Research Fellowship (https://www.jsps.go.jp/english/e-pd/index.html, JP19J14097). These funders had no role in study design, data collection and analysis, decision to publish, or preparation of the manuscript.

**Competing interests:** The authors have declared that no competing interests exist.

intrinsic growth capacity and regenerate. One mechanism that contributes to this process is the activation of receptors by growth factors that promote the growth capacity of neurons [2]. In neurons, intrinsic mechanisms that mediate regeneration are conserved across species. The nematode *Caenorhabditis elegans* is a useful experimental model for elucidating axon regeneration mechanisms [3]. Genetic screens have identified several signaling molecules required for axon regeneration [4–7]. Recent studies have identified the JNK MAP kinase (MAPK) signaling pathway as a key intrinsic regulator of regeneration initiation in *C. elegans* [8,9]. The pathway is activated by a signaling cascade involving the mammalian HGF-like growth factor, SVH-1, and its receptor tyrosine kinase (RTK), SVH-2 [5]. Thus, SVH-1–SVH-2 signaling is essential for neuronal regeneration but not for neuronal development.

SVH-1 contains a Kringle domain, an N-terminal domain related to a plasminogen activation peptide, and a serine protease domain [5,10]. These protein domains are also present in mammalian HGF and plasminogen, suggesting that SVH-1 is a member of the HGF/plasminogen family. SVH-2 is homologous to c-Met, a mammalian HGF RTK [5]. The sequence homology of SVH-1 to HGF and SVH-2 to c-Met suggests that SVH-1–SVH-2 functions as a ligand–receptor pair in axon regeneration (Fig 1A). In mammals, HGF acts as a growth factor for c-Met and induces the activation of complex intracellular signaling pathways [11]. However, unlike plasminogen, HGF lacks the Asp–His–Ser triad, which is essential for catalytic activity and cannot function as a protease (Fig 1A; [11]). In contrast, plasminogen possesses an intact catalytic triad and functions as a protease. Protease activity is essential for plasminogen to remodel components of the extracellular matrix (ECM) (Fig 1A; [12]). Thus, amino acids in the catalytic triad determine whether the protein functions as a protease or growth factor. SVH-1, similar to plasminogen, has an intact catalytic triad. However, its protease activity is not required for SVH-1 to activate SVH-2, as with c-Met-mediated HGF signaling [5]. Furthermore, SVH-1 regulates larval growth in a protease activity-dependent manner, independent of SVH-2 [10]. The *svh-1* gene is the only *C. elegans* gene encoding a protein of the HGF/plasminogen family. Therefore, SVH-1 has a dual-function: it acts as an HGF-like growth factor to promote axon regeneration and as a plasminogen-like protease required for larval development (Fig 1A; [10]). However, it is unclear how SVH-1 can function as a growth factor despite having an intact catalytic triad. Therefore, there must be a switch mechanism that converts SVH-1 from a protease to a growth factor to function in axon regeneration (Fig 1B).

Here, we attempted to identify the factor in regulating the functional conversion of SVH-1 during axon regeneration. We have genetically identified several genes (*svh*) involved in the JNK signaling pathway that regulates axon regeneration [5], including the *svh-17*/*cdd-2* gene encoding cytidine deaminase (CDD). In this study, we investigated the relationship between *cdd-2* and *svh-1* in the regulation of axon regeneration, and show that CDD-2 is required for axon regeneration by converting SVH-1 from a protease to a growth factor.

## Results

### SVH-17/CDD-2 is required for axon regeneration

Recent genetic studies have shown that the JNK MAPK pathway regulates axon regeneration in *C. elegans* [8]. The JNK cascade is inactivated by the MAPK phosphatase VHP-1, and loss-of-function mutations in *vhp-1* lead to hyper-activation of the JNK pathway, resulting in developmental arrest at an early larval stage [13]. To identify factors involved in JNK signaling, a genome-wide RNAi screen was performed to isolate suppressors of *vhp-1* lethality [5]. Ninety-two RNAi clones were isolated and named *svh* genes for suppressors of *vhp-1* lethality. The *svh-17* gene is identical to the *cdd-2* gene encoding CDD (Fig 2A). CDD-2 contains a motif for the zinc-binding region of CDDs (Fig 2A; [14]). Phylogenetic analysis using the catalytic

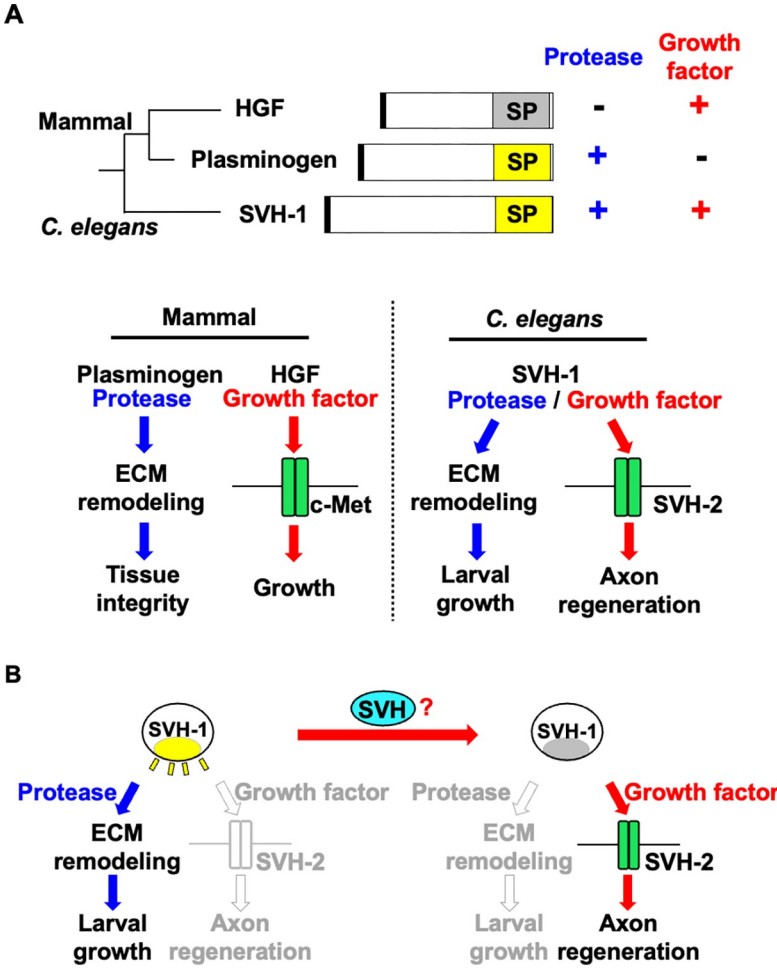

**Fig 1. Functions of SVH-1 in larval growth and axon regeneration.** (A) Relationship between mammalian HGF/plasminogen and *C. elegans* SVH-1. SVH-1 is a dual-function protein that acts as a plasminogen-like protease and an HGF-like growth factor. SP: serine protease-like domain. (B) How is SVH-1 converted from a protease-active to a growth factor-active form to function in axon regeneration?

domain of the enzymes shows that CDD-2 forms a cluster with cytosine nucleoside deaminase [14].

To investigate whether *cdd-2* is involved in axon regeneration, we generated a mutant *cdd-2 (km70)* lacking the initiation codon using the CRISPR/Cas9 system (S1A Fig). Previous studies have shown that RNAi knockout of *cdd-2* reduced viability [14], but the *cdd-2(km70)* mutant had no effect on viability or development (S2A Fig). We examined the regeneration of laser-severed axons in γ-aminobutyric acid (GABA)-releasing D-type motor neurons (Fig 2B). In young adult wild-type (WT) animals, approximately 70% of axons initiated regeneration within 24 h after injury (Fig 2B and 2C, and S1 Table), whereas the frequency of axon regeneration was reduced in *cdd-2(km70)* mutant animals (Fig 2B and 2C, and S1 Table). To confirm that the axon regeneration defect was due to the *cdd-2* mutation, we constructed a *Pcdd-2::cdd-2* (genomic DNA) transgene, an approximately 1.4 kb genomic DNA fragment containing the entire coding region of *cdd-2* and its promoter (S3A Fig). The introduction of the *Pcdd-2::cdd-2* (genomic DNA) transgene as an extrachromosomal array significantly rescued the defect associated with the *cdd-2(km70)* mutation (Fig 2C and S1 Table). Another *cdd* gene, *cdd-1*, is

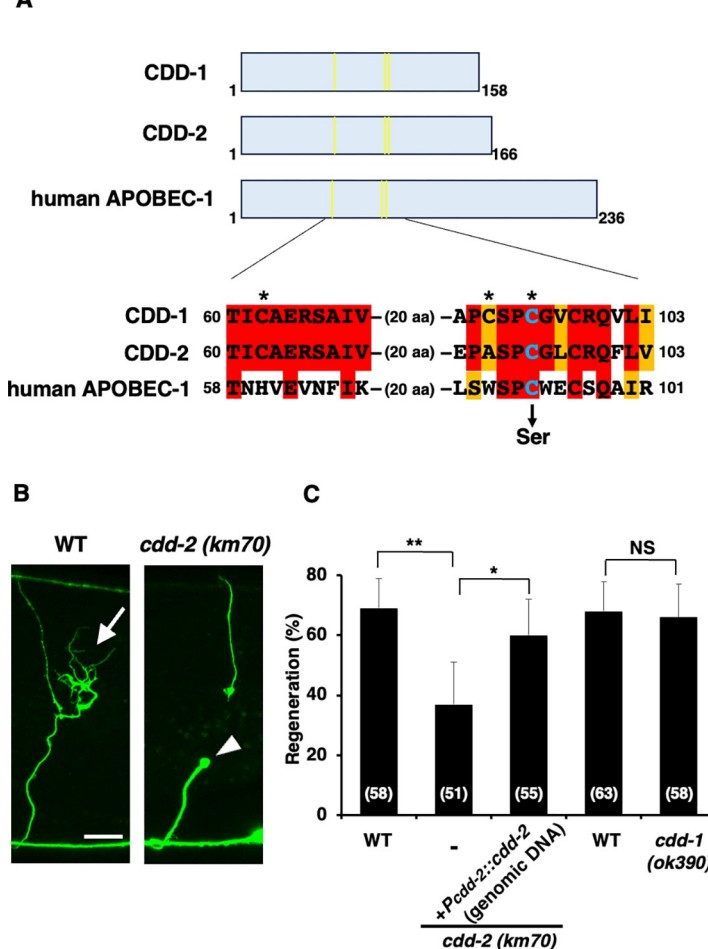

**Fig 2. CDD-2 is required for axon regeneration.** (A) Domain structure of CDD-1, CDD-2 and human APOBEC-1. Zinc-binding domains are shown in yellow. The active site alignment is shown. Identical and similar residues are highlighted with red and orange shading, respectively. Zinc-binding amino acids are indicated by asterisks. The conserved cysteine residue (blue) required for CDD activity is shown. (B) Representative D-type motor neurons in WT and *cdd-2(km70)* mutant animals 24 h after laser surgery. In WT animals, a severed axon has regenerated a growth cone (arrow). In mutants, the proximal ends of the axons failed to regenerate (arrowhead). Scale bar = 10 μm. (C) Percentages of axons that initiated regeneration 24 h after laser surgery. The number of axons examined (n) is shown. Error bars indicate 95% CIs. *$P < 0.05$; **$P < 0.01$ as determined by the Chi-square test and corrected by the false discovery rate (FDR) controlling Benjamini–Hochberg (BH) procedure for multiple comparisons. NS, not significant.

present in *C. elegans* (Fig 2A; [14]). However, in *cdd-1(ok390)* mutants (S1B Fig), axons regenerated normally after axon injury (Fig 2C and S1 Table), suggesting that CDD-2 plays a specific role in axon regeneration after laser axotomy.

## CDD-2 regulates axon regeneration via SVH-1

Since our RNAi screen for *svh* genes was originally designed to identify components that function in the JNK pathway [5], we investigated where CDD-2 functions in this pathway. Previously, we have demonstrated that SVH-2 c-Met-like RTK, which is activated by the SVH-1 HGF-like growth factor, mediates the activation of the JNK cascade in axon regeneration (Fig 3A; [5]). We examined the relationship between CDD-2 and the SVH-1–SVH-2 signaling pathway. We found that *svh-2* overexpression in D-type motor neurons under the *unc-25*

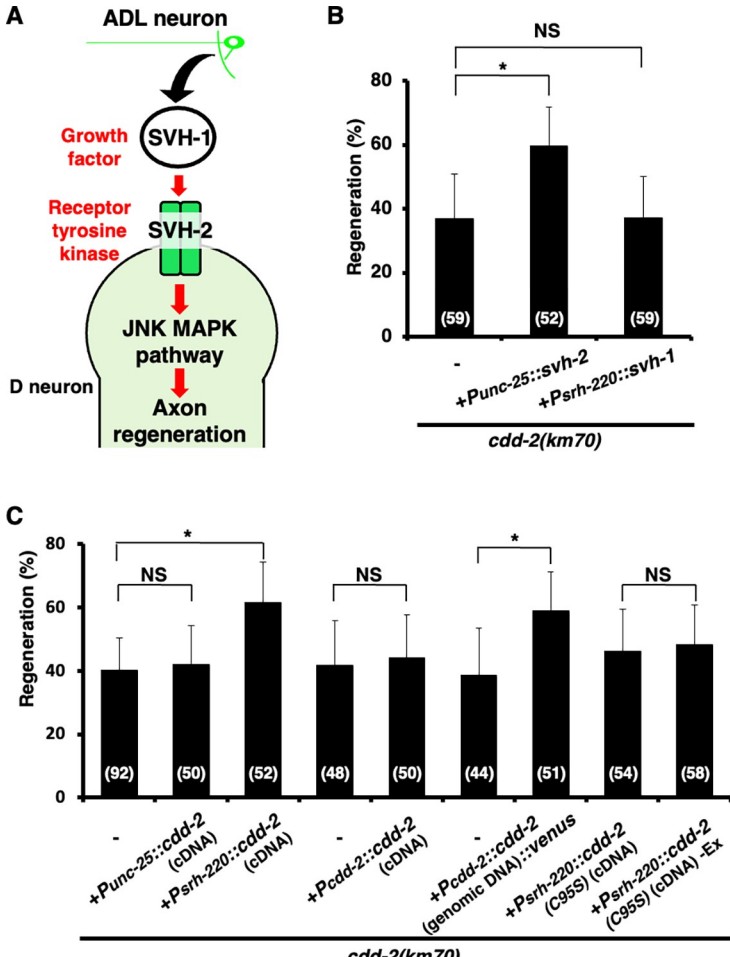

**Fig 3. CDD-2 functions in the SVH-1–SVH-2 signaling pathway to regulate axon regeneration.** (A) SVH-1–SVH-2 signaling pathway required for axon regeneration in *C. elegans*. SVH-1 is constitutively expressed and secreted by ADL sensory neurons in the head. SVH-2 is an RTK activated by SVH-1. Activated SVH-2 activates the JNK pathway. (B and C) Percentages of axons that initiated regeneration 24 h after laser surgery. The number of axons examined (n) is shown. Error bars indicate 95% CIs. *$P < 0.05$ as determined by the Chi-square test and corrected by the false discovery rate (FDR) controlling Benjamini–Hochberg (BH) procedure for multiple comparisons. NS, not significant.

promoter suppressed the regeneration defect in *cdd-2(km70)* mutants, whereas *svh-1* overexpression under the control of the ADL neuronal promoter *srh-220* failed to suppress the *cdd-2* defect (Fig 3B and S1 Table). These results suggest that CDD-2 functions downstream of SVH-1 and upstream of SVH-2 in axon regeneration. The *cdd-2(km70)* defect was not rescued by *cdd-2* (cDNA) expression in D-type motor neurons using the *unc-25* promoter (Figs 3C and S3A, and S1 Table), suggesting that CDD-2 does not function in injured D-type neurons. Since *svh-1* is expressed in ADL neurons [5], we examined whether CDD-2 functions in ADL neurons. We found that *cdd-2* (cDNA) expression under the *srh-220* promoter in ADL neurons was able to rescue the *cdd-2(km70)* defect in axon regeneration (Figs 3C and S3A, and S1 Table). Thus, the role of CDD-2 in regeneration is required in the ADL sensory neuron, the same cell where SVH-1 is required, suggesting that CDD-2 acts on SVH-1.

Next, we tested whether *cdd-2* is expressed in ADL neurons. Thompson et al. [14] analyzed the expression pattern of *cdd-2* using two reporter genes: a transcriptional fusion type containing only the upstream sequence and a translational fusion type. They showed that both

reporter genes were expressed in the intestine but not in ADL neurons. The expression of *cdd-2* (cDNA) from its own promoter and 3′-untranslated region (UTR) in *cdd-2(km70)* mutants did not rescue the axon regeneration defect (Figs 3C and S3A, and S1 Table). However, the defect was rescued when genomic DNA, but not *cdd-2* cDNA, was expressed from the *cdd-2* promoter and 3′-UTR or when *cdd-2* (cDNA) was expressed from the *srh-220* promoter and *unc-54* 3′-UTR (Figs 2C, 3C and S3A, and S1 Table). Thus, the intronic region of the *cdd-2* genomic DNA is required to properly regulate *cdd-2* transcription from its promoter. Accordingly, to monitor *cdd-2* expression, we constructed a reporter gene, *Pcdd-2::cdd-2* (genomic DNA)::*venus*, by fusing *cdd-2* (genomic DNA) containing the promoter and the entire *cdd-2* gene to the *venus* gene (S3A Fig). Indeed, we confirmed that the *Pcdd-2::cdd-2* (genomic DNA)::*venus* reporter gene was functional in axon regeneration (Fig 3C and S1 Table). Using this construct, we found that the *cdd-2* gene is expressed in ADL neurons (S3B Fig).

## CDD-2 converts the function of SVH-1 from a protease to a growth factor by modifying *svh-1* mRNA

CDD belongs to a family of enzymes that catalyze the deamination of cytidine (C) to uridine (U). The editing of *Apolipoprotein-B* (*ApoB*) mRNA is a well-known example of C-to-U RNA-editing in mammals [15]. This editing is mediated by the *ApoB* mRNA-editing CDD subunit-1 (APOBEC-1), which converts a specific C-residue in *ApoB* mRNA to a U. This deamination generates an in-frame premature stop codon, resulting in ApoB48, a short isoform of ApoB. The full-length form (ApoB100) and the ApoB48 isoform regulate lipid metabolism [16]. Based on this precedent, CDD-2 may change the function of SVH-1 from a protease to a growth factor by editing *svh-1* mRNA. To confirm that CDD-2 acts as a CDD, we examined an enzyme-dead mutant of CDD-2 in axon regeneration. The *ApoB* mRNA-editing activity of APOBEC-1 depends on its Cys-93 residue (Fig 2A; [17]). CDD-2 contains a conserved Cys-95 site that corresponds to mammalian Cys-93 (Fig 2A). We created the *cdd-2(C95S)* mutant, in which Cys-95 is substituted with serine (Fig 2A). The expression of *cdd-2(C95S)* in *cdd-2 (km70)* mutants using the *srh-220* promoter did not restore the axon regeneration defect (Fig 3C and S1 Table). Therefore, the enzymatic activity of CDD-2 is required for axon regeneration after laser ablation.

The mammalian growth factor HGF has a serine protease fold but lacks the essential Asp–His–Ser catalytic triad found in all active serine proteases. Instead, it has an Asp–Gln–Tyr triad (Fig 4A; [10]), suggesting that the amino acid Gln or Tyr in the triad may determine that HGF acts as a growth factor. SVH-1 possesses the essential catalytic triad, with the codons His-755 (C̲AU) and Ser-898 (U̲CC) in the Asp–His–Ser catalytic triad of SVH-1 containing C-residues (Fig 4A). When these C-residues are edited to U-residues in *svh-1* mRNA, the codons for His-755 (C̲AU) and Ser-898 (U̲CC) are replaced by tyrosine (U̲AU) and phenylalanine (UU̲C/U̲) codons, respectively (Fig 4A). If CDD-2 is involved in editing *svh-1* mRNA, it is expected that *H755Y* or *S898F* mutation would mimic its effect and bypass the requirement for CDD-2 in axon regeneration. When *svh-1(H755Y)* or *svh-1(S898F)* was expressed from the *srh-220* promoter in *cdd-2(km70)* mutants, we found that *H755Y*, but not *S898F*, was able to suppress the *cdd-2(km70)* defect (Fig 4B and S1 Table), suggesting that CDD-2 mediates C-to-U editing of *svh-1* mRNA, replacing His-755 at the protease catalytic site with tyrosine and producing protease-inactive and growth factor-active SVH-1. The Ser-898 residue is located within the Asp–His–Ser catalytic triad (Fig 4A) and is essential for protease activity. However, the function of SVH-1(S898F) in axon regeneration still depends on CDD-2, suggesting that the His-755 residue determines whether SVH-1 acts as a protease or a growth factor. Therefore, SVH-

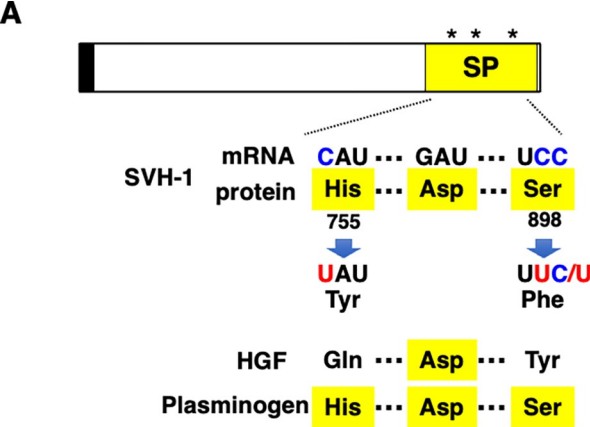

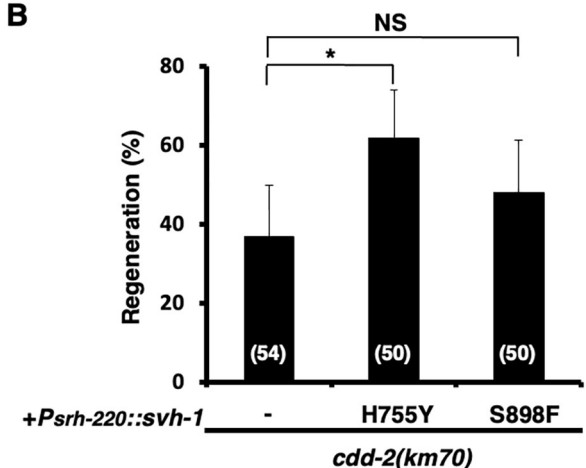

**Fig 4. CDD-2 converts the function of SVH-1 from a protease to a growth factor to promote axon regeneration.**
(A) Comparison of the serine protease-like domains of SVH-1, HGF, and plasminogen. Domains are shown as follows: signal peptide (black box) and serine protease-like domain (SP). Codon changes His-755 to Tyr and Ser-898 to Phe are shown. The catalytic triad Asp–His–Ser is indicated by asterisks. (B) Percentages of axons that initiated regeneration 24 h after laser surgery. The number of axons examined (n) is shown. Error bars indicate 95% CIs. *$P < 0.05$ as determined by the Chi-square test and corrected by the false discovery rate (FDR) controlling Benjamini–Hochberg (BH) procedure for multiple comparisons. NS, not significant.

1 in the form of His-755 cannot activate SVH-2, and His-755 must be converted to a tyrosine for SVH-1 to be active as a growth factor.

## A secondary structure around the C-residue in the His-755 codon of *svh-1* mRNA is required for axon regeneration

The recognition of RNA-editing sites is proposed to depend on the secondary structure of the substrate RNA [18]. Analysis of the *svh-1* mRNA sequence using the CentroidFold program [19] identified a stem-loop secondary structure in which the C-residue of the His-755 codon is localized in a loop (Fig 5A). To test the importance of the stem-loop structure for SVH-1 function, we introduced mutations in *svh-1*, *svh-1(s-mut)*, which disrupt the stem structure (6 out of 9 bp) but do not alter the resulting amino acid sequence (Fig 5A). The *svh-1* deletion mutation *ok2531* was used to evaluate the effects of *svh-1(s-mut)* on axon regeneration and larval development. The originally isolated strain with the *svh-1(ok2531)* mutation has an extra copy

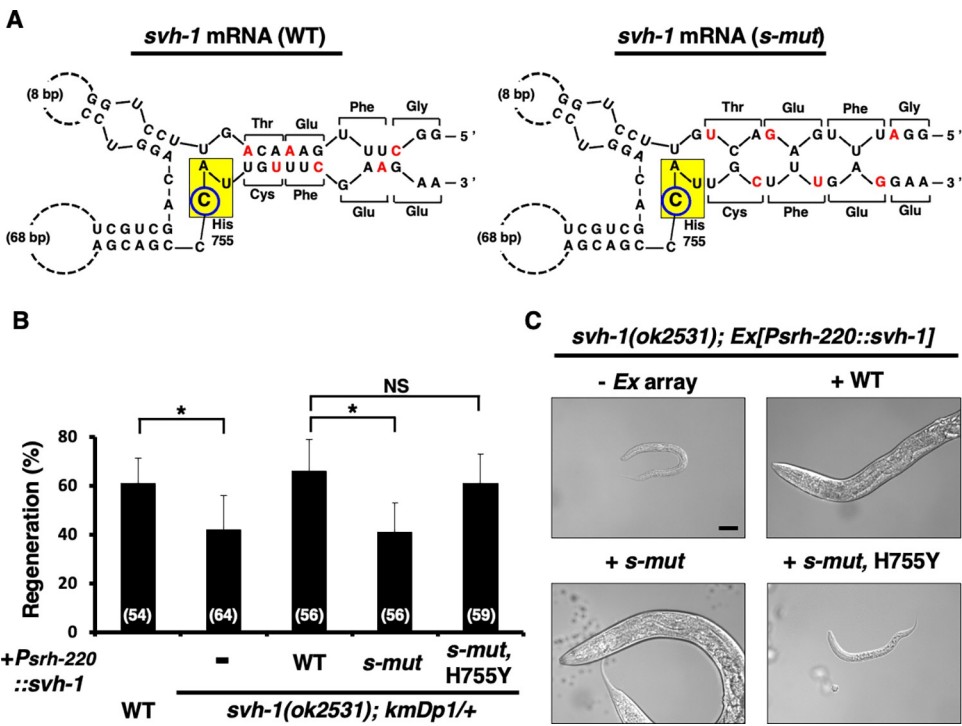

**Fig 5. Stem-loop structure of *svh-1* mRNA.** (A) Computer modeling of the 400 nucleotide *svh-1* mRNA using the CentroidFold program. The C-residue for His-755 is circled in blue. Base pair structure, mutation sites (red letters) and encoded amino acids are shown. The *svh-1(s-mut)* mutation is shown on the right part. (B) Percentages of axons that initiated regeneration 24 h after laser surgery. The number of axons examined (n) is shown. Error bars indicate 95% CIs. *$P < 0.05$ as determined by the Chi-square test and corrected by the false discovery rate (FDR) controlling Benjamini–Hochberg (BH) procedure for multiple comparisons. NS: not significant. (C) Light microscopy of animals grown for 5 days. Anterior is to the left. Scale bar = 50 μm.

of the *svh-1* gene fragment (*kmDp1*) translocated to the right end of the LGX (S4 Fig), which shows defects in axon regeneration but does not affect development [10]. When this duplicated *kmDp1* is removed from the *svh-1(ok2531)*; *kmDp1/+* strain, the resulting mutant *svh-1 (ok2531)* stops growing at the L1 larval stage [10]. As observed previously [5], WT *svh-1* expression from the *srh-220* promoter in *svh-1(ok2531); kmDp1/+* mutants was able to rescue the axon regeneration defect, whereas *svh-1(s-mut)* expression could not (Fig 5B and S1 Table). However, the *svh-1(s-mut)* mutation was able to rescue the L1 arrest phenotype observed in *svh-1(ok2531)* deletion mutants (Fig 5C). Thus, the disruption of the stem-loop structure around His-755 does not affect the protease activity of SVH-1 but prevents its function as a growth factor. Therefore, we constructed the *svh-1(s-mut, H755Y)* mutation in which His-755 in *svh-1(s-mut)* was changed to tyrosine. We found that the *svh-1(s-mut, H755Y)* mutation was able to rescue the axon regeneration defect in *svh-1(ok2531)* but not the larval developmental defect (Fig 5B and 5C, and S1 Table), suggesting that the stem-loop structure of *svh-1* mRNA is required for the replacement of the His-755 (CAU) codon with a tyrosine (UAU) codon.

## The *cdd-2* gene expression in ADL neurons correlates with the timing required for axon regeneration

SVH-1 has two functions: protease activity for larval development at the L1 stage and a growth factor for axon regeneration at the L4 to adult stages (Fig 6A; [10]). Since SVH-1 functions in

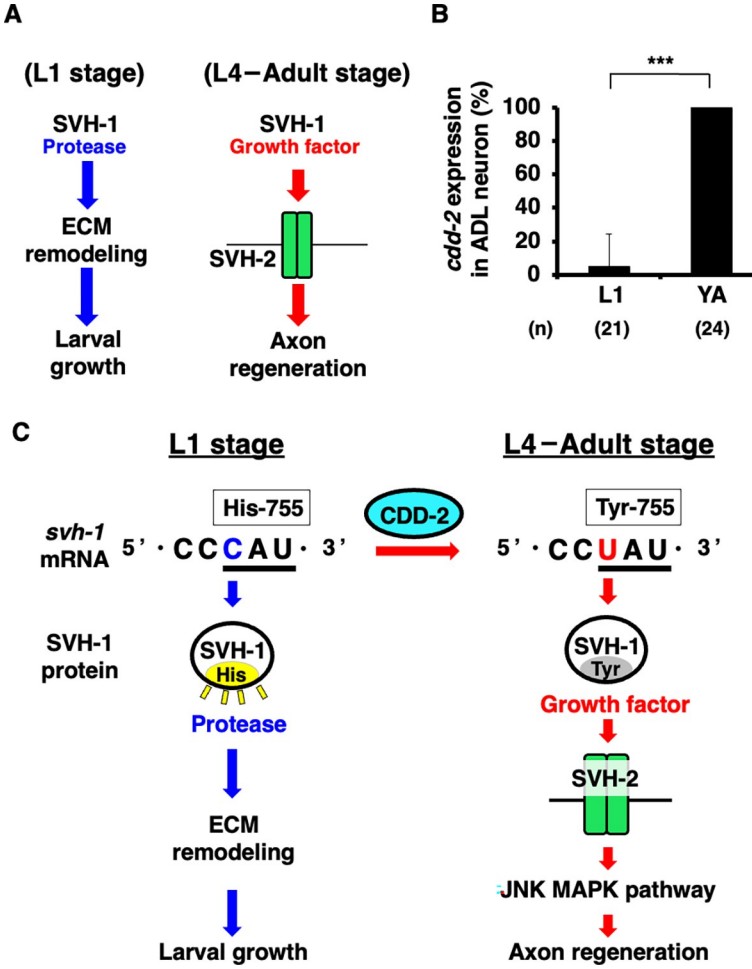

**Fig 6. Timing of *cdd-2* gene expression in ADL neurons.** (A) Stage-specific functions of SVH-1. SVH-1 has two functions, a protease for larval development at the L1 stage and a growth factor for axon regeneration at the L4–adult stage. (B) Percentages of animals expressing *cdd-2*. The numbers (n) of axons examined are shown. Error bars indicate 95% CIs. \*\*\**P*<0.001 as determined by Welch's t-test. (C) Schematic diagram of the modulation of SVH-1 functions regulated by CDD-2.

ADL neurons and its protease activity is required for larval development, inactivation of SVH-1 protease activity by CDD-2 should not work at the L1 stage. Therefore, the *cdd-2* gene is expected to be expressed specifically in ADL neurons at the L4 to adult stages. This possibility was assessed by looking at the expression pattern of the *Pcdd-2*::*cdd-2* (genomic DNA)::*venus* reporter gene during development. We found that *cdd-2* expression was detected at the L4–young adult stages but not at the L1 stage (Fig 6B). This expression pattern is consistent with RT-PCR analysis results, which showed higher *cdd-2* expression in adult mRNA [14], suggesting that during the L4–young adult stages, CDD-2 converts SVH-1 into a growth factor that is required for axon regeneration.

If *cdd-2* expression alone were sufficient to replace His-755 in *svh-1* mRNA with tyrosine, forced *cdd-2* expression in the L1 stage would cause larval arrest because the SVH-1 protein with Tyr-755 instead of His-755 loses protease activity. However, *cdd-2* overexpression from the *srh-220* promoter in WT animals did not affect larval development (S2B Fig), suggesting that *cdd-2* expression alone does not cause editing of *svh-1* mRNA. Further events, such as the activation of CDD-2 by axonal injury, may be necessary.

## Discussion

In this study, we demonstrate that CDD-2 converts the function of SVH-1 from a protease to a growth factor. Our genetic results suggest that CDD-2 deaminates the C of *svh-1* mRNA to U, replacing His-755 (C̱AU) with tyrosine (U̱AU) in the Asp–His–Ser catalytic triad, producing a protease-inactive and growth factor-active SVH-1. Thus, the Tyr-755 form of SVH-1 loses protease activity but gains the ability to act as a growth factor. There are two possible explanations for these results. First, the activation of SVH-2 by SVH-1 requires the inactivation of the protease-active SVH-1. Second, the His-755 form of SVH-1 does not function as a growth factor. Although the *svh-1(S898F)* mutant loses protease activity, it fails to suppress the *cdd-2* defect. Thus, the protease-inactive SVH-1(S898F) can function as a growth factor only when His-755 is substituted with Tyr-755 in *svh-1* mRNA by CDD-2, suggesting that SVH-1 with His-755 cannot interact with or activate SVH-2. Therefore, CDD-2 specifically changes the His-755 of SVH-1 to a motif that can activate SVH-2.

In mammals, APOBEC-1 is a CDD responsible for deaminating C within *ApoB* mRNA [15]. This RNA-editing process converts a specific glutamine codon (C̱AA) into a premature termination codon (U̱AA), resulting in a C-terminal truncated ApoB48 protein. In this way, APOBEC-1 enables the synthesis of full-length (ApoB100) and short isoforms (ApoB48) of ApoB from a single mRNA [15]. *ApoB* mRNA is expressed in the liver and small intestine, whereas APOBEC-1 is only expressed in the small intestine. Consequently, ApoB48 and ApoB100 are produced in the small intestine and liver, respectively. Tissue-specific editing of *ApoB* mRNA is an important factor in lipid metabolism [20]. However, in contrast to this scenario, the protease-active and growth factor-active isoforms of SVH-1 act in the same cell (ADL neuron) to regulate larval development during the L1 stage and axon regeneration during the L4–adult stages, respectively [5,10]. Therefore, CDD-2-mediated inactivation of the SVH-1 protease activity should not occur during the L1 stage; *cdd-2* is only expressed at a later, distinct developmental stage. Thus, *cdd-2* expression at a specific developmental stage is essential for the functional conversion of SVH-1 from a protease to a growth factor during axon regeneration. Forced *cdd-2* expression at the L1 stage does not lead to the editing of *svh-1* mRNA, suggesting that the enzymatic activity of CDD-2 may be activated by axon injury.

The cytidine editing site in *ApoB* mRNA is located within a stem-loop secondary structure [21], which is important for RNA recognition by editing factors [22]. Similarly, in *svh-1* mRNA, the His-755 encoding CAU codon is within a stem-loop secondary structure. Our genetic analysis suggests that the stem-loop structure of *svh-1* mRNA is required for SVH-1 function in axon regeneration. We propose that this secondary structural component of *svh-1* mRNA is recognized by CDD-2. APOBEC-1 has general AU-binding activity [22], and CDD-2 also possesses RNA-binding activity with an affinity for AU-rich RNA templates [14]. However, no biochemical evidence exists that CDD-2 catalyzes the editing of *svh-1* mRNA in animals. If CDD-2 activity is activated by axon injury, detecting *svh-1* mRNA-editing within animals would be more difficult. In this scenario, axon injury to D-type motor neurons should induce CDD-2-dependent *svh-1* mRNA-editing. Nevertheless, both *svh-1* and *cdd-2* are expressed and function in ADL neurons, which are spatially distant from D-type neurons. Consequently, the question remains as to how SVH-1 in ADL neurons is transported to D neurons. Interestingly, recent biochemical analyses have revealed that both *svh-1* mRNA and CDD-2 protein are entrapped in extracellular vesicles (EVs) [23,24]. Furthermore, although the *svh-1* reporter gene with the *nls::venus*-fused *svh-1* promoter demonstrated expression exclusively in ADL neurons [5], data from the CeNGEN project [25] indicated that transcripts of *svh-1* were detected in other neurons in addition to ADL neurons. These results raise the possibility that EVs containing *svh-1* mRNA and CDD-2 are delivered from ADL neurons to

D neurons. Upon axon injury, CDD-2-dependent editing and subsequent translation of *svh-1* mRNA occurs, resulting in the production of SVH-1(H755Y), a protein with growth factor activity. Therefore, it is necessary to elucidate the regulatory mechanisms governing *svh-1* mRNA-editing by CDD-2.

## Materials and methods

### *C. elegans* strains

The *C. elegans* strains used in this study are listed in S2 Table. All strains were maintained on nematode growth medium plates and fed with bacteria of the OP50 strain, as described previously [26].

### Plasmids and primers

The *Pcdd-2::cdd-2* (genomic DNA) clone was generated by amplification of approximately 1.4 kb of the *cdd-2* gene from genomic DNA by PCR (using the cdd-2genF and cdd-2genR) and insertion into the TOPO vector (Invitrogen). The *cdd-2* (cDNA) was generated by oligonucleotide-directed PCR using a pACT *C. elegans* cDNA library [27] as a template and verified by DNA sequencing. *Psrh-220::cdd-2* (cDNA) was replaced by the *svh-1* cDNA of *Psrh-220::svh-1* with the *cdd-2* cDNA. *Punc-25::cdd-2* (cDNA) and *Pcdd-2::cdd-2* (cDNA) were made by inserting the *cdd-2* cDNA fragment into the pSC325 vector and the *Pcdd-2::cdd-2* (genomic DNA) plasmid, respectively, and were both verified by DNA sequencing. *Psrh-220::cdd-2(C95S)* (cDNA), *Psrh-220::svh-1(H755Y)*, *Psrh-220::svh-1(S898F)*, *Psrh-220::svh-1(s-mut)*, and *Psrh-220::svh-1(s-mut, H755Y)* were generated by oligonucleotide-directed PCR using *Psrh-220::cdd-2* (cDNA) or *Psrh-220::svh-1* as a template and verified by DNA sequencing. *Pcdd-2::cdd-2* (genomic DNA)::*venus* was constructed by inserting a VENUS fragment into the *Pcdd-2::cdd-2* (genomic DNA) plasmid and verified by DNA sequencing. The *pU6::cdd-2_sgRNA* plasmid was made by replacing the *unc-119* target sequence of *pU6::unc-119_sgRNA* plasmid (Addgene) with 5′-atcttgaggaagactattcg-3′, corresponding to the genomic sequence within the *cdd-2* gene. *Punc-25::svh-2*, *Psrh-220::svh-1*, *Psrh-220::cfp*, *Pofm-1::gfp*, *Pmyo-2::dsred-monomer*, and *Peft-3::cas9-sv40_nls::tbb-2 3′-UTR* have been described previously [5,10,28,29]. The oligonucleotide primers used in this study are listed in S3 Table.

### Generation of the *cdd-2* mutation using CRISPR/Cas9

The *cdd-2(km70)* deletion mutant was generated using the CRISPR/Cas9 system, as described previously [29]. The *pU6::cdd-2_sgRNA* plasmid (160 ng/µl) was co-injected together with the *Peft-3::cas9-sv40_nls::tbb-2 3′-UTR* (30 ng/µl) and *Pmyo-2::dsred-monomer* (20 ng/µl) plasmids into the KU501 strain. An F1 animal carrying the transgene was picked, and genomic DNA from its descendants was amplified by PCR and digested with *Aci*I [30] to detect short insertions or deletions in the *cdd-2* gene. The descendants of these animals were selected to obtain the *cdd-2* homozygous mutant. The *cdd-2(km70)* mutant contains duplicated DNA regions that include fragments of the *cdd-2* gene. One of these regions has a 41 bp deletion that lacks the initiation codon for *cdd-2*. The other carries a 3 bp in-frame deletion that removes the fourth Asn of CDD-2.

### Transgenic animals

Transgenic animals were obtained by the standard *C. elegans* microinjection method [31]. The *Pcdd-2::cdd-2* (genomic DNA) (25 ng/µl), *Punc-25::cdd-2* (cDNA) (25 ng/µl), *Psrh-220::cdd-2* (cDNA) (17 ng/µl), *Psrh-220::cdd-2(C95S)* (cDNA) (25 ng/µl), *Pcdd-2::cdd-2* (cDNA) (25 ng/

μl), *Pcdd-2::cdd-2* (genomic DNA)::*venus* (25 ng/μl), *Punc-25::svh-2* (25 ng/μl), *Psrh-220::svh-1* (25 ng/μl), *Psrh-220::svh-1(H755Y)* (25 ng/μl), *Psrh-220::svh-1(S898F)* (25 ng/μl), *Psrh-220:: svh-1(s-mut)* (25 ng/μl), *Psrh-220::svh-1(s-mut, H755Y)* (25 ng/μl), *Psrh-220::cfp* (25 ng/μl), *Pofm-1::gfp* (25 ng/μl), and *Pmyo-2::dsred-monomer* (25 ng/μl) plasmids were used in *kmEx1482 [Pcdd-2::cdd-2* (genomic DNA) + *Pmyo-2::dsred-monomer], kmEx1483 [Punc-25:: cdd-2* (cDNA) + *Pmyo-2::dsred-monomer], kmEx1484* and *kmEx1627 [Psrh-220::cdd-2* (cDNA) + *Pmyo-2::dsred-monomer], kmEx1499* and *kmEx1628 [Psrh-220::cdd-2(C95S)* (cDNA) + *Pmyo-2::dsred-monomer], kmEx1489 [Pcdd-2::cdd-2* (cDNA) + *Pmyo-2::dsred-monomer], kmEx1490 [Pcdd-2::cdd-2* (genomic DNA)::*venus* + *Pmyo-2::dsred-monomer], kmEx1485 [Punc-25::svh-2* + *Pmyo-2::dsred-monomer], kmEx1486 [Psrh-220::svh-1* + *Pmyo-2:: dsred-monomer], kmEx1487 [Psrh-220::svh-1(H755Y)* + *Pmyo-2::dsred-monomer], kmEx1488 [Psrh-220::svh-1(S898F)* + *Pmyo-2::dsred-monomer], kmEx1493 [Psrh-220::svh-1(s-mut)* + *Pmyo-2::dsred-monomer], kmEx1494 [Psrh-220::svh-1(s-mut, H755Y)* + *Pmyo-2::dsred-mono-mer],* and *kmEx1491 [Pcdd-2::cdd-2* (genomic DNA)::*venus* + *Psrh-220::cfp* + *Pofm-1::gfp],* respectively. For each injection, pBluescript II plasmid DNA was added as required to reach a final concentration of 200 ng/μl of total DNA in the injection solution.

## Axotomy

Axotomy was performed as described previously [5]. Young adult hermaphrodite animals were immobilized with 0.7% sodium azide or 20 mM levamisole solution in M9 buffer on a 2% agarose pad under a cover slip. D-type motor neurons expressing GFP were imaged with a fluorescence microscope. Selected D-type neurons were severed using a 440 nm MicroPoint Ablation Laser System from Photonic Instruments. The animals were transferred to an agar plate and remounted for fluorescent imaging ~24 h after surgery. Axons that grew in length by 5 μm or more were scored as "regenerated." Proximal axon segments that showed no change after 24 h were counted as "no regeneration." At least 20 animals with 1–3 axotomized commissures were observed for most experiments.

## Microscopy

Standard fluorescent images of transgenic animals were observed under ×20 or ×100 objective of a Nikon ECLIPSE E800 fluorescent microscope and photographed with a Zyla CCD camera. Confocal fluorescent images were taken on a Zeiss LSM-800 confocal laser scanning microscope with a ×63 objective.

## Prediction of the secondary structure of *svh-1* mRNA

To predict the secondary structure of the region surrounding the codon for His-755 in *svh-1* mRNA (2,100–2,499 bp), the CentroidFold program (http://rtools.cbrc.jp/centroidfold/)) with McCaskill(BL) or CONTRAfold parameters was used.

## Statistical analysis

Statistical analyses were carried out using the GraphPad program (http://www. graphpad.com/ quickcalcs/). The modified Wald method (https://www.graphpad.com/quickcalcs/ confInterval1/) was employed to calculate 95% confidence intervals (CIs), while the Chi-square test (http://www. graphpad.com/quickcalcs/contingency1/) was used to calculate two-tailed *P*-values. In order to control the false discovery rates (FDR) in multiple comparisons, the Benjamini–Hochberg (BH) procedure [32] was applied using the False Discovery Rate Online

Calculator (https://tools.carbocation.com/FDR). Welch's t-test was performed using a t-test calculator (http://www.graphpad.com/quickcalcs/ttest1/).

## Supporting information

**S1 Fig. Genome structures.** (A and B) Genome structures of the *svh-17*/*cdd-2* (A) and *cdd-1* (B) genes. Exons are indicated by boxes, introns and untranslated regions by bars. The domain shown is the zinc-binding region (yellow). The bold line below indicates the extent of the deletion region in the *km70* mutant. Nucleotides and corresponding amino acids around the deleted region are also indicated. The *cdd-2(km70)* mutant contains duplicated DNA regions that include fragments of the *cdd-2* gene. One of these regions has a 41 bp deletion that lacks the initiation codon for *cdd-2*. The other carries a 3 bp in-frame deletion that removes the fourth Asn of CDD-2. The *cdd-1(ok390)* mutant harbors a deletion of 898 bp and an insertion of 7 bp, removing the zinc-binding region.
(TIF)

**S2 Fig. Light microscopy of animals.** (A) WT and *cdd-2(km70)* mutant animals grown for 4 days are shown. Scale bar = 50 µm. (B) WT animals carrying *Psrh-220*::*cdd-2* (cDNA) or *Psrh-220*::*cdd-2(C95S)* (cDNA) grown for 4 days are shown. Scale bar = 50 µm.
(TIF)

**S3 Fig. Expression of the *cdd-2* gene in ADL neurons.** (A) Genomic structures of fusion genes. Exons are indicated by boxes, introns and untranslated regions by bars. (B) Expression of the *Pcdd-2*::*cdd-2* (genomic DNA)::*venus* gene in ADL neurons. Fluorescent images of animals carrying *Pcdd-2*::*cdd-2* (genomic DNA)::*venus* at young adult stage are shown. ADL neurons (arrowheads) are visualized by CFP under the control of the *srh-220* promoter. Anterior is to the left. Schematic diagram of ADL neurons in the head is shown. Scale bar = 10 µm.
(TIF)

**S4 Fig. Genome structure of the *svh-1* gene in *svh-1(ok2531)*; *kmDp1/+* strain.** The *svh-1 (ok2531)*; *kmDp1/+* strain has the *svh-1(ok2531)* mutation and an extra copy of the *svh-1* gene fragment (*kmDp1*) translocated to the right end of the LGX.
(TIF)

**S1 Table. Raw data of genotypes tested by axotomy.**
(XLSX)

**S2 Table. Strains used in this study.**
(XLSX)

**S3 Table. Primers used in this study.**
(XLSX)

## Acknowledgments

We thank Dr. Yoshiki Sakai for his helpful discussion. and the *Caenorhabditis* Genetic Center (CGC), National Bio-Resource Project and *C. elegans* Knockout Consortium for materials. Some strains were provided by the CGC, funded by the NIH Office of Research Infrastructure Programs (P40 OD10440).

## Author Contributions

**Conceptualization:** Tatsuhiro Shimizu, Kunihiro Matsumoto, Naoki Hisamoto.

**Data curation:** Tatsuhiro Shimizu, Takafumi Nomachi, Naoki Hisamoto.

**Formal analysis:** Tatsuhiro Shimizu, Takafumi Nomachi, Naoki Hisamoto.

**Funding acquisition:** Tatsuhiro Shimizu, Kunihiro Matsumoto, Naoki Hisamoto.

**Investigation:** Kunihiro Matsumoto, Naoki Hisamoto.

**Methodology:** Tatsuhiro Shimizu, Naoki Hisamoto.

**Project administration:** Kunihiro Matsumoto, Naoki Hisamoto.

**Resources:** Naoki Hisamoto.

**Supervision:** Kunihiro Matsumoto, Naoki Hisamoto.

**Validation:** Tatsuhiro Shimizu, Kunihiro Matsumoto, Naoki Hisamoto.

**Visualization:** Tatsuhiro Shimizu, Naoki Hisamoto.

**Writing – original draft:** Kunihiro Matsumoto.

**Writing – review & editing:** Tatsuhiro Shimizu, Kunihiro Matsumoto, Naoki Hisamoto.

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
