## [Decision Letter · Decision Letter 0]

9 Feb 2024

Dear Dr Matsumoto,

Thank you very much for submitting your Research Article entitled 'A cytidine deaminase regulates axon regeneration by modulating the protease function of the Caenorhabditis elegans HGF/plasminogen family protein SVH-1' to PLOS Genetics.

The manuscript was fully evaluated at the editorial level and by independent peer reviewers. The reviewers appreciated the attention to an important topic but identified some concerns that we ask you address in a revised manuscript. Some of these concerns would ideally be addressed by additional experiments, but many can be addressed by editing of the manuscript or data analysis. Also, as Academic Editor, I ask that you make the following changes in a revised manuscript

- Increase detail in reporting methods and reagents. This includes a) fully describing the genotype of *juIs76* in the first entry row in Table S2 and fully describe the balancer/duplications in this table, b) listing any primers used for plasmid construction and sequence of reagents used for CRIPSR genome editing, and c) concentrations of other plasmids or DNA in microinjection pools used for transgenesis.

- Comply with the PLOS Genetics Data policy by providing a full data table for experimental results or increase clarity in data presentation. Does Table S1 contain all experimental results? If so, the reuse of data in different panels/figures within the manuscript needs to be made clear and explicit. Readers expect that independent determinations underlie different panels (like 2C and 5B for WT); if different panels in this manuscript use the same biological replicates/determinations, any data reuse must be explicitly stated in every figure legend, in the Methods (under Statistical analysis), and in the manuscript text. 

We therefore ask you to modify the manuscript according to the review and editor recommendations. Your revisions should address all of the points made by each reviewer. or stated above.

Yours sincerely,

Anne C. Hart

Academic Editor

PLOS Genetics

Gregory P. Copenhaver

Editor-in-Chief

PLOS Genetics

Reviewer's Responses to Questions

**Comments to the Authors:**

Reviewer #1: Shimizu et al. presented the switching mechanism of SVH-1 signaling by the deamination of cytidine (C) to uridine (U) processed with CDD-2. Based on the well-designed genetic analysis of the axon regeneration phenotype of adult C. elegans, the authors uncovered the responsible protease inactivation mechanism, the site of the action, and the timing of the event. Particularly the readers of PLoS Genetics would have great interest in the suggested mechanism of mRNA editing by CDD-2 in a specific location and timing. Logically constructed experiments and results support the proposed mechanisms. Overall the manuscript is beautifully composed with a shrewd logical flow although the discussion part needs an improvement for discriminating the hypothesis derived from the literature and what is derived from the current study. Also, there are some concerns before publishing the manuscript. The manuscript should be improved to strengthen the proposed claims by answering the following comments and questions.

1. The main question is somewhat vague. The keywords like protease activity, RNA editing, switching mechanism, and axon regeneration seem equally discussed and in what context the authors try to discover the signaling mechanism is unclear. From the abstract, it seems that the authors are trying to emphasize the finding of the switching mechanism of protease activity, but in what context it is important is not enough explained. Does the switching mechanism of protease activity possibly contribute to the biological process other than the axon regeneration of C. elegans? Or, what is the physiological meaning of the protease activity-based switching mechanism for axon regeneration and larval growth? Given that the RNA editing mechanisms for switching a protein function are already reported, clarifying the difference between the reported studies and the current study in the context of biological importance should be necessary. For example, what is the difference from the following study in terms of the scientific focus?

Song, Yuanquan, et al. "Regulation of axon regeneration by the RNA repair and splicing pathway." Nature neuroscience 18.6 (2015): 817-825.

2. What triggers the SVH-2? I am wondering whether the inactive form of the protease is enough to activate SVH-2 or not? In other words, is there any specific motif to activate SVH-2, and does the modification by CDD-2 specifically replace the protease active form with the exact motif for activating SVH-2?

3. Does the laser surgery for the L1 larvae inhibit axon regeneration? If the proposed hypothesis is true, L1 larvae should have a different phenotype compared to the adults. Most probably, because L1 has basically active regeneration properties, the laser surgery of L1 still shows active regeneration same as the adults. Still, using overexpression or genetic manipulation can show the different phenotypes of L1 larvae for the laser axotomy.

4. ADL neuron is located at the head ganglion while the D neuron is at the body. How is the inactivated SVH-1 at the ADL transported to the D neuron?

5. On line146, the authors conducted the overexpression experiments with the ADL-specific promotor. I am wondering whether the expression of SVH-1 with the intrinsic promotor of SVH-1 suppresses the cdd-2 mutant defect or not. Such experiments clarify the possibility of the involvement of the non-ADL SVH-1 in axon regeneration.

6. The way of statistical analysis is very tricky or even awkward. To my knowledge, Fisher’s exact test determines independence or association between two categorical variables. When a contingency table is given, one p-value will be calculated by the test. The authors somehow calculate multiple p-values for Figs 2, 3, 4, and 5 as if Fisher’s test is used for the comparison among the multiple groups to determine whether the groups are statistically different. Although I believe that the way of statistical tests do not affect the conclusion in this case, at least the authors should clarify the precise protocol and the validity for the statistical analysis since the authors unusually use Fisher’s exact test. Also, the issue of multiple comparisons for statistical tests might be involved in this case.

7. Most of the evidence for the hypothesis is shown indirectly. Ideally, direct monitoring for the mRNA editing or protease activity will support the proposed hypothesis greatly. Is it possible to detect the deaminated mRNA sequence by RNAseq? Probably the mRNA from different sources such as the intestine interferes with the target processed mRNA, but still, the comparison between cdd-2 mutant and wildtype, or L1 and adult should be different if the proposed hypothesis is true. At least, including the limitation of the current study in the discussion section will clarify the meaning of the current research and provide the basis for future studies.

8. Since the beginning of the discussion section starts with the literature/hypothesis review and the result of the current research starts from the second paragraph, the structure of the discussion part looks unbalanced. Probably the current second paragraph starting from line 271 should be moved to the beginning of the discussion section. Starting with the interpretation of the current result consolidates a basis for continuing discussions. Maybe, the current first paragraph of the discussion section should be merged into the introduction section since it motivates the current research.

Reviewer #2: The manuscript by Shimizu et al explored an interesting question: how SVH-1 switches from a protease functioning in development to a protease-inactive HGF-like growth factor that facilitates axon regeneration. Their previous screen has identified SVH-17/CDD-2 as a regulator of axon regeneration. Here they found that CDD-2 regulates axon regeneration through SVH-1. As CDD-2 is an enzyme that catalyze the deamination of cytidine to uridine, they hypothesize that CDD-2 converts SVH-1 from a protease to a non-protease by C-to-U RNA editing. Indeed, they found that transgenic expression of a C-to-U mutation of SVH-1 was sufficient to rescue defects in cdd-2 mutant. They further demonstrated that the stem-loop structure of svh-1 mRNA is critical for CDD-2 to recognize and/or catalyze its substrate. The study leveraged a series of clever genetic mutations and identified an interesting regulation mechanism. I am supportive for its publication in Plos Genetics. However, some concerns need to be addressed before it can be published:

1. They found that SVH-1 overexpression was not able to rescue the regeneration defects in cdd-2 mutants, while SVH-2 overexpression was able to rescue. It’s not convincing that overexpression of SVH-2 was sufficient to transduce the signal to promote axon regeneration in the absence of ligand (non-protease variant of SVH-1), unless a constitutively active SVH-2 was used.

2. The authors argued that the timing of cdd-2 gene expression is critical for its role in editing svh-1 mRNA for promoting axon regeneration. In addition to showing cdd-2 expression pattern, this conclusion needs to be further confirmed by assessing axon regeneration in cdd-2 mutants at different stages (e.g. L3). If cdd-2 mutant shows no defects in axon regeneration at L3 stage, it would support that cdd-2 function in axon regeneration is time-specific. Also, time-specific transgenic expression of CDD-2 or SVH-1 variants through heat-shock promoter can help to further support their conclusion.

3. Line 132, data on axon regeneration in ccd-1(ok390) are missing in Fig. 2C.

Reviewer #3: Comments on Shimizu et al.

Shimizu et al. reported novel regulation via RNA editing of the dual functions of SVH-1 on axon regeneration and development. They show that the cytosine deaminase CDD-2 alters a key codon in the SVH-1 mRNA that blocks SVH-1 protease function. A stem-loop in the SVH-1 mRNA is critical for this editing step. This editing allows SVH-1 to switch its function from early development, when it acts via its protease to enable growth, to axon regeneration, when it acts in a protease-independent manner via the receptor SVH-2.

Overall this is an elegant study that elucidates a novel and interesting mechanism. A few key experiments and some expanded discussion would strengthen the conclusions.

1. To prove CDD-2 acts as cytosine nucleoside deaminase, it would be nice to test an enzyme-dead mutant of CDD-2 in axon regeneration.

2. Regarding the Asp-His-Ser catalytic triad, if all three amino acids must be present for SVH-1 protease activity, then any inactive SVH-1 mutation should rescue cdd-2 mutant phenotype. But in Figure 4B, H755Y and S898F behave differently. This argues that loss of protease activity is not sufficient for bypassing cdd-2. Biochemical experiments showing the amount of activity present in each mutant would be ideal (and also assessing binding to the receptor). Alternatively, the overall model should at least be revised.

3. Authors reported CDD-2 expression levels are low in L1 but increased in L4. So in L1, SVH-1 would mostly be protease-active form. However, L1 animals have been shown to regenerate well. This should be discussed.

4. Also regarding the timing of expression, the model predicts that overexpression CDD-2 under srh-220 promoter should inactive SVH-1 early in development,

---

## [Decision Letter · Decision Letter 1]

20 May 2024

Dear Dr Matsumoto,

Thank you very much for submitting your Research Article entitled 'A cytidine deaminase regulates axon regeneration by modulating the functions of the Caenorhabditis elegans HGF/plasminogen family protein SVH-1' to PLOS Genetics.

The manuscript was fully evaluated at the editorial level and by independent peer reviewers. The reviewers appreciated the attention to an important topic and almost all of their concerns have been addressed. Reviewer 1 had two comments on the revised manuscript. The first askd for text revisions to increase clarity. We do not require any revisions to address this comment; this is optional. The second comment addresses statistical analysis. We leave the decision regarding Fisher's versus other comparisons testing to the authors, but require that the Methods section explain the logic behind this choice. However, the reviewers' comment regarding multiple comparisons testing must be addressed because the same control samples were used repeatedly in multiple figures in the manuscript. The manuscript's citation of a publication arguing this is unnecessary for clinical trials is not persuasive. The standard approach in the field is to use multiple comparisions testing and, if a revised manuscript is submitted, this statistical approach must be used. 

Yours sincerely,

Anne C. Hart

Academic Editor

PLOS Genetics

Gregory P. Copenhaver

Section Editor

PLOS Genetics

Reviewer's Responses to Questions

**Comments to the Authors:**

Reviewer #1: The authors' responses mostly satisfied the concerns I suggested. The only two things that I strongly suggest improving are:

1. The manuscript still omits the description of the limitations of the current study and sometimes it makes the audience skeptical. The previous comment No. 4 pointed out the question about how spatial signal transportation occurs and it would be beneficial to discuss about this in the discussion section.

Also, related to the previous comment No.5, the CENGEN results show the expression of svh-1 in the other neurons. Together with the fact that ref. 5 only uses the NLS-fused expression of svh-1, a clear description of the intention of the experiment and the limitation of the design should make a solid story.

2. Related to the previous comment No. 6, it is highly suggested to consult with a good statistician. Clearly, the problem of the multi-comparison in this study is not the case in Ref. 30. The paper discusses the different layers of the problem that are clearly shown in the title. And if we closely look into the reference about the adjustment in multiple-testing as shown in Ref.30 (the following paper), this is a case study of the clinical research and no theoretical validation is described.

10. Wason J.M.S., Stecher L., Mander A.P. Correcting for multiple-testing in multi-arm trials: is it necessary and is it done? Trials. 2014;15(1):364.

The experiments of mutant and rescue lines are highly related and can not be regarded as distinct tests.

Moreover, the main point of the previous comment was the proper use of Fisher's exact test. It is still doubtful to be inappropriate use of the method.

Reviewer #2: The revised manuscript has adequately addressed the review critiques. I am happy to support its publication.

Reviewer #3: The authors have adequately addressed issues raised by the reviewers. The revised manuscript presents very interesting data and is suitable for publication.

**Have all data underlying the figures and results presented in the manuscript been provided?**

Reviewer #1: Yes

Reviewer #2: Yes

Reviewer #3: Yes

PLOS authors have the option to publish the peer review history of their article (what does this mean?). If published, this will include your full peer review and any attached files.

Reviewer #1: No

Reviewer #2: No

Reviewer #3: No

---

## [Editor Report · Decision Letter 2]

11 Jul 2024

Dear Dr Matsumoto,

We are pleased to inform you that your manuscript entitled "A cytidine deaminase regulates axon regeneration by modulating the functions of the Caenorhabditis elegans HGF/plasminogen family protein SVH-1" has been editorially accepted for publication in PLOS Genetics. Congratulations!

Yours sincerely,

Anne C. Hart

Academic Editor

PLOS Genetics

Gregory P. Copenhaver

Section Editor

PLOS Genetics

Comments from the reviewers (if applicable):

**Data Deposition**

http://datadryad.org/submit?journalID=pgenetics&manu=PGENETICS-D-23-01297R2

**Press Queries**

---

## [Editor Report · Acceptance letter]

23 Jul 2024

PGENETICS-D-23-01297R2 

A cytidine deaminase regulates axon regeneration by modulating the functions of the *Caenorhabditis elegans* HGF/plasminogen family protein SVH-1 

Dear Dr Matsumoto, 

We are pleased to inform you that your manuscript entitled "A cytidine deaminase regulates axon regeneration by modulating the functions of the *Caenorhabditis elegans* HGF/plasminogen family protein SVH-1" has been formally accepted for publication in PLOS Genetics! Your manuscript is now with our production department and you will be notified of the publication date in due course.

With kind regards,

Zsofia Freund

PLOS Genetics

On behalf of:
